# Genome-Wide Identification and Evolution Analysis of the *CYP76* Subfamily in Rice (*Oryza sativa*)

**DOI:** 10.3390/ijms24108522

**Published:** 2023-05-10

**Authors:** Mingao Zhou, Yifei Jiang, Xuhui Liu, Weilong Kong, Chenhao Zhang, Jian Yang, Simin Ke, Yangsheng Li

**Affiliations:** 1State Key Laboratory of Hybrid Rice, College of Life Sciences, Wuhan University, Wuhan 430072, China; 2021202040077@whu.edu.cn (M.Z.); 2021102040039@whu.edu.cn (Y.J.); 2020202040079@whu.edu.cn (X.L.); weilong.kong@whu.edu.cn (W.K.); zch_nx@126.com (C.Z.); 2020202040075@whu.edu.cn (J.Y.); 2021202040080@whu.edu.cn (S.K.); 2Shenzhen Branch, Genome Analysis Laboratory of the Ministry of Agriculture, Agricultural Genomics Institute at Shenzhen, Chinese Academy of Agricultural Sciences, Shenzhen 518120, China; 3Department of Biomedical Informatics, School of Basic Medical Science, Peking University Health Science Center, Beijing 100191, China

**Keywords:** abiotic stress, cytochrome P450 protein, expression patterns, gene family, *Oryza genus*

## Abstract

The *CYP76* subfamily, a member of the *CYP* superfamily, plays crucial roles in the biosynthesis of phytohormones in plants, involving biosynthesis of secondary metabolites, hormone signaling, and response to environmental stresses. Here, we conducted a genome-wide analysis of the *CYP76* subfamily in seven AA genome species: *Oryza sativa* ssp. *japonica*, *Oryza sativa* ssp. *indica, Oryza rufipogon*, *Oryza glaberrima*, *Oryza meridionalis*, *Oryza barthii*, and *Oryza glumaepatula.* These were identified and classified into three groups, and it was found that Group 1 contained the largest number of members. Analysis of cis-acting elements revealed a large number of elements related to jasmonic acid and light response. The gene duplication analysis revealed that the *CYP76* subfamily expanded mainly in SD/WGD and tandem forms and underwent strong purifying selection during evolution. Expression pattern analysis of *OsCYP76s* in various developmental stages revealed that the majority of *OsCYP76s* exhibit relatively restricted expression patterns in leaves and roots. We further analyzed the expression of CYP76s in *O. sativa*, *japonica*, and *O. sativa*, *indica* under cold, flooding, drought, and salt abiotic stresses by qRT-PCR. We found that *OsCYP76-11* showed a huge increase in relative expression after drought and salt stresses. After flooding stress, *OsiCYP76-4* showed a greater increase in expression compared to other genes. CYP76 in *japonica* and *indica* showed different response patterns to the same abiotic stresses, revealing functional divergence in the gene family during evolution; these may be the key genes responsible for the differences in tolerance to indica japonica. Our results provide valuable insights into the functional diversity and evolutionary history of the *CYP76* subfamily and pave the way for the development of new strategies for improving stress tolerance and agronomic traits in rice.

## 1. Introduction

Cytochrome P450 (CYP) is a ubiquitous and ancient enzyme protein found in plants, animals, fungi, bacteria, and viruses [1,2]. The CYP superfamily is the largest enzymatic family in plants and accounts for approximately 1% of all protein-coding genes [3]. CYP enzymes are involved in a wide range of biological processes, including biosynthesis of primary and secondary metabolites, hormone signaling, and response to environmental stresses [4,5,6,7,8,9]. The CYP76 subfamily, a member of the CYP superfamily, plays a crucial role in the biosynthesis of phytohormones, such as jasmonic acid, and secondary metabolites, such as alkaloids, terpenoids, and flavonoids, in plants [10,11,12,13].

In recent years, research on the members of the CYP76 subfamily has become increasingly popular due to its vital role in plant development and adaptation to environmental stresses [14,15,16]. In Arabidopsis, *CYP76C2* was found to respond to leaf senescence and cell culture aging; various processes that lead to cell death. [17]. Knockout mutant rice with *CYP76C6* overexpression (OEs) and CRISPR/Cas9 were generated by genetic transformation and gene editing techniques, and *CYP76C6* was found to play a key role in the degradation of IPU. [18]. *CYP76M7* was reported to have catalyzed the insertion of oxygen at the C11α position and was involved in the production of antifungal phytocassanes [19]. Moreover, the CYP76 subfamily is involved in the synthesis of several secondary metabolites with potential applications in the pharmaceutical and agricultural industries [10,20,21]. The geraniol 8-hydroxylase *CYP76B6* from *C. roseus* plays a key regulatory role in terpenoid indole alkaloid biosynthesis and, in 2001, was found to belong to the CYP76 family [10]. Additionally, another CYP76 enzyme (*CYP76A26*) from *C. roseus* with iridoid oxidase activity, could form 7-deoxyloganetic acid for the biosynthesis of secoiridoids and terpene indole alkaloids through catalyzing the 3-step oxidation of iridodial-nepetalactol [22]. *CYP76C4* from *Arabidopsis thaliana* could convert geraniol into 8-hydroxy- or 9-geraniol [11]. *CYP76M5*, *CYP76M6*, and *CYP76M8* from *Oryza sativa* were all shown to carry out C-7β hydroxylation and may act consecutively in oryzalexin biosynthesis [23]. Based on the important role of CYP76s, understanding the functions and regulation of this gene family is crucial for the development of new strategies for plant breeding and crop improvement.

The CYP76 subfamily has been extensively studied in *Arabidopsis thaliana* and several other plant species [24]. However, there is limited information available on the functional characterization and evolutionary history of this gene family in rice (*Oryza sativa*), one of the most important crops worldwide [25,26]. In recent years, the availability of the complete genome sequence of rice has facilitated the identification and functional analysis of CYP76 genes.

In this study, we aim to identify and analyze the CYP76 subfamily in seven AA genome species, including *O. sativa*, japonica (Osj), *O. sativa*, indica (Osi), *O. rufipogon* (Or), *O. glumaepatula* (Oglu), *O. glaberrima* (Ogla), *O. meridionalis* (Om), and *O. barthii* (Ob). Our goal is to explore their evolutionary history and functional diversity. To achieve this, we conducted a comprehensive genome-wide analysis to identify all members of the CYP76 subfamily in *Oryza*, followed by a rigorous phylogenetic analysis to infer their evolutionary relationships. We then delved into the sequence and protein properties of CYP76 genes and studied their chromosomal localization, orthogroup identification, and collinearity. Moreover, we examined the expression patterns of these genes in different tissues and developmental stages and under various abiotic stresses in *O. sativa*, japonica and *O. sativa*, indica. The results of our investigation could provide valuable insights into the functional diversity and evolutionary history of the CYP76 subfamily in rice, which will be instrumental in elucidating the biosynthesis of phytohormones and secondary metabolites. Furthermore, our findings may pave the way for the development of new strategies for improving stress tolerance and agronomic traits in rice.

## 2. Results

### 2.1. Identification of CYP76 Genes in Oryza Genus

In this study, a total of 122 nonredundant CYP76 genes were identified from the seven studied AA genome rice, with 17 in *O. sativa*, japonica, 21 in *O. sativa*, indica, 21 in *O. rufipogon*, 17 in *O. glumaepatula*, 18 in *O. glaberrima*, 14 in *O. meridionalis*, and 14 in *O. barthii* (Appendix A). Obviously, *O. sativa*, indica and *O. rufipogon* contained more CYP76 genes than the other studied species. Interestingly, we observed that the distribution of these genes was uneven across the chromosomes, with all CYP76 genes being absent on chromosomes 6, 7, and 11 in all studied species (Figure 1a). Moreover, CYP76 genes were only present on chromosome 12 in *O. sativa*, indica and *O. rufipogon*, and were absent on chromosome 6 in *O. glumaepatula.* Additionally, the distribution of CYP76 genes was very similar in the seven species and the CYP76 genes on chromosome 10 were all distributed proximally, suggesting that the physical location of some CYP76 genes was conserved during the evolution of *Oryza* plants.

To conduct a comprehensive analysis of the physical and chemical properties of CYP76 proteins, physicochemical properties of these proteins were predicted, with the number of amino acid residues ranging from 443 (ORUFI10G03760.1) to 1042 (OBART09G12160.1), molecular weight from 49,816 to 113,182, and isoelectric points from 6.1 (OMERI08G17560.1) to 9.85 (ORUFI08G07480.1). Additionally, most CYP76 proteins (80%) have an isoelectric point greater than seven, suggesting that CYP76 proteins may be basic proteins. To explore where CYP76 proteins exert their functions, the subcellular localization of these proteins was predicted, with most proteins having subcellular localization sites in chloroplast-like vesicle membranes, while others were located in mitochondria, endoplasmic reticulum, cytoplasm, and plasma membranes, suggesting that they may be involved in photosynthesis and chloroplast metabolism (Appendix A).

To further explore the number of differences and developmental relationships of CYP76 genes among these species, phylogenetic tree, group classification, and orthologous group identification were conducted. Our results showed that all CYP76 genes were classified into three groups, with Group 1 containing the largest number of genes (Figure 1b). Additionally, we identified twelve orthologous groups using OrthoFinder (Oxford, United Kingdom), with Or 1 possessing the largest number of genes (Appendix A).

### 2.2. Gene Structure, Conserved Motif, and Domain Analysis

In this study, we utilized the MEME online tool to identify 20 conserved motifs among 122 CYP76 proteins (Appendix A). Our analysis revealed that all CYP76 proteins possessed a similar composition, number, and arrangement of motifs and domains, which were classified into three distinct groups (Figure 2). Notably, we found that Motif 2 and Motif 10 were distributed at the N-terminus of all proteins, while Motifs 3, 9, and 13 were distributed at the C-terminus of all proteins (Figure 2b). Interestingly, Motif 16 and Motif 19 were exclusively present in Group 1 and Group 2, respectively, while Motif 17 and Motif 18 were absent in Group 1. These observations suggest that these motifs may play distinct roles in various physiological processes. Furthermore, all proteins in the three groups contained two primary conserved domains, namely, the p450 and CypX superfamily (Figure 2c). Our gene structure analysis indicated that *OsCYP76s* had relatively fewer introns, with the number ranging from 1 to 5 (Figure 2d). As expected, genes of the same group shared similar exon/intron structures. Most members of Group 1 and Group 2 possessed short intron structures, in contrast to Group 3, which had longer introns. These findings provide new insights into the functional diversity of CYP76 gene family members in *Oryza* species.

### 2.3. Cis-Elements Analysis of CYP76s

To investigate the potential roles of cis-elements in CYP76 genes, we conducted an analysis of the 2 kb upstream sequences of the transcription start codon of these genes (Appendix A). Our findings, generated by using the PlantCare database, revealed a total of 18 cis-elements, in addition to two conventional promoter elements, TATA-box and CAAT-box, which can be classified into three categories: abiotic stress-related elements, development-related elements, and hormone response-related elements (Figure 3).

Abiotic stress-related elements were abundant in the CYP76 genes promoter region, with anaerobic induction (ARE), defense and stress responsive (TC-rich repeats), dehydration and salt responsive (DRE), drought inducibility (MBS), and low-temperature responsive (LTR and ABRE) elements accounting for 35.86, 5.96, 23.26, 18.37, and 16.45% of the total, respectively. These results suggest that CYP76 genes may play a role in plant responses to cold, drought, and anaerobic stress. Hormone response-related elements were also abundant in the CYP76 genes promoter region, with ABA (ABRE) and MeJA (TGACG and CGTCA motif) accounting for 30.56 and 29.22% of the total, respectively, while IAA (TGA-box and AuxRR-core) and GA (GARE-motif, P, and TATC box) both accounted for 14.75%. The development related elements represented the largest category of cis-elements identified in the CYP76 genes promoter region, with light response (Box 4, chs-CMA1a, and G box), meristem expression (CAT-box), zein metabolism regulation (O^2^-site), circadian control (circadian), seed specific regulation (RY-element), endosperm expression (GCN4_motif), and flavonoid biosynthetic genes regulation (MBSI) elements accounting for 51, 16.37, 14.58, 6.14, 4.35, 2.81, and 0.77% of the total, respectively. Interestingly, seed-specific regulatory elements were mainly distributed in the promoters of Group 1 and Group 2 members, suggesting that these groups may play a role in the regulation of seed development. Taken together, these results suggest that CYP76 genes in plants may be involved in a range of physiological, biochemical, and developmental processes (Figure 3c).

### 2.4. Collinearity and Selective Forces Analysis of CYP76s

In order to investigate the expansion patterns of the CYP76 gene family, we conducted an analysis of collinear duplication events among seven *Oryza* species using the MCScanX with default method (Appendix A). Our findings, which were visualized using TBtools (Guangzhou, China), revealed the presence of 2, 1, 1, 1, 2, 1, and 1 collinear gene pairs in *O. sativa*, *japonica*, *O. sativa*, *indica*, *O. rufipogon*, *O. glumaepatula*, *O. glaberrima*, *O. meridionalis*, and *O. barthii*, respectively (Figure 4). Interestingly, most of the collinear genes were distributed on chromosome 8 and 9. Furthermore, all duplication gene pairs were found to be of the SD/WGD or tandem duplication type, and the gene pairs showed similar positions in these species, indicating that the expansion patterns of the CYP76 gene family were roughly similar. In these seven species, the Ka/Ks values of all duplication gene pairs ranged from 0.30 to 0.67, which is less than one, suggesting that these genes have undergone strong purifying selection during evolution. Interestingly, the SD/WGD duplication events were estimated to occur between 49.01 and 88.64 Mya, while most tandem duplication events were estimated to occur between 11.64 and 28.37 Mya, indicating that gene pairs in SD/WGD duplication events were conserved during evolution (Table 1).

### 2.5. Expression Pattern Analysis of OsCYP76 Genes and RNA-seq Analysis under Abiotic Stresses

To investigate the expression patterns of *OsCYP76s* in various developmental stages, we obtained datasets from the RiceXPro database (Appendix A). Our analysis revealed that the majority of *OsCYP76* genes exhibit relatively restricted expression patterns in leaves and roots (Figure 5a). Interestingly, collinear gene pairs display distinct expression patterns. For instance, *OsCYP76-15* (*LOC_Os10g08230*) was under-expressed in all tissues, whereas *OsCYP76-6 (LOC_Os03g14400)* showed broad expression across diverse tissues and developmental processes. *OsCYP76-12* displayed relatively narrow expression patterns in all tissues, while the expression levels of *OsCYP76-9* were higher in leaves and roots, suggesting functional differentiation among these genes after the duplication event. Furthermore, *OsCYP76-12* and *OsCYP76-15* were both under-expressed in nearly all tissues, indicating that certain CYP76 gene family members are functionally redundant. Notably, *OsCYP76-11* was exclusively expressing during endosperm development, indicating its potential crucial role in this process. Additionally, *OsCYP76-1*, *OsCYP76-10*, and *OsCYP76-13* in Group 1 were solely expressed in roots, suggesting their unique functions in root development.

Previous studies have reported the significant roles of the CYP76 gene family in response to abiotic stresses. Therefore, we investigated the expression patterns of *OsCYP76s* in shoot and root under salt, drought, cold, and flood stress conditions using RNA-seq data from TENOR (Appendix A). RNA-seq results revealed that *OsCYP76s* in roots and leaves exhibited differential responses to the same abiotic stresses. In the shoot, almost all *OsCYP76* genes exhibited altered expression levels in response to cold, flooding, drought, and salt stresses. Specifically, following exposure to cold, drought, and flooding stresses, half of the genes exhibited decreased expression levels, while the other half exhibited increased expression levels. Following salt stress, most genes exhibited increased expression levels, except for *OsCYP76-14*, which exhibited decreased expression levels, and *OsCYP76-7* and *OsCYP76-11*, which did not show significant changes in expression (Figure 5b). In the roots, the majority of *OsCYP76* genes displayed decreased expression levels after exposure to cold and flood stress, while half of them responded to salt stress, and almost all of them exhibited altered expression levels in response to drought stress, with half of them showing increased expression levels and the other half showing decreased expression levels (Figure 5c).

### 2.6. CYP76 in O. sativa, Japonica and O. sativa, Indica Were Shown Different Responses to Abiotic Stresses through qRT-PCR Analysis

To validate the expression differences of the CYP76 gene family in *O. sativa*, japonica and *O. sativa*, indica, we conducted an analysis of CYP76 gene expression in leaves and roots under four abiotic stress conditions through qRT-PCR (Figure 6). After flooding treatment, we found that more than half of CYP76 genes increased in leaves of *japonica*, while almost all of them decreased in leaves of indica (Figure 6a,c). It is worth noting that *OsiCYP76-4* showed a greater increase in expression compared to other genes. Additionally, most CYP76 genes showed little difference in relative expression in *japonica* roots, whereas in indica, most showed increased relative expression (Figure 6b,d). After salt and drought treatment, we observed that the expression of almost all CYP76 genes was increased in japonica leaves, whereas in indica leaves, half of the genes were up-regulated and the other half were down-regulated (Figure 6a,c). Notably, *OsCYP76-11* showed a huge increase in relative expression after drought and salt stress, suggesting that *OsCYP76-11* may play an important function in drought and salt stress. In roots of japonica and indica, almost all CYP76 genes showed increased relative expression (Figure 6b,d). After cold treatment, we observed an increase in CYP76 genes expression in leaves of both japonica and indica, while the expression levels in roots differed significantly between the two subspecies (Figure 6). Specifically, there was little difference in CYP76 genes expression in roots of japonica before and after treatment, whereas in indica, most CYP76 genes exhibited increased expression (Figure 6b,d). These findings suggest that the expression patterns of the CYP76 gene family in response to abiotic stresses differ between japonica and indica, indicating functional differentiation within the gene family during evolution.

## 3. Discussion

In recent years, the availability of the complete genome sequence of rice has facilitated the identification and functional analysis of various gene families, such as *GH3*, *AGC*, and *GAox* [27,28,29]. The CYP76 gene family is a member of the larger cytochrome P450 superfamily and plays important roles in various biological processes in plants, including biosynthesis of primary and secondary metabolites, hormone signaling, and response to environmental stresses [10,11,12,13]. Eight members of the CYP76 subfamily were identified in *Arabidopsis thaliana* and a detailed analysis of these members was performed [24]. However, the information on the CYP76 subfamily in rice was not a comprehensive systematic analysis. In this study, we completely identified and analyzed the CYP76 subfamily in seven AA genome *Oryza* genus, explored their evolutionary history, and investigated their functional diversity and expression patterns under various abiotic stresses.

Our genome-wide analysis identified a total of 122 CYP76 genes in the seven *Oryza* genus, with 17 in *O. sativa*, japonica, 21 in *O. sativa*, indica, 21 in *O. rufipogon*, 17 in *O. glumaepatula*, 18 in *O. glaberrima*, 14 in *O. meridionalis*, and 14 in *O. barthii* (Appendix A). Obviously, the more evolutionarily related *O. sativa*, *indica* and *O. rufipogon* contained the largest number of CYP76 genes. These CYP76 numbers are roughly twice as many as those in *Arabidopsis* [24]. We also found that different *Oryza* species have different numbers of CYP76 genes, suggesting that the CYP76 subfamily has undergone lineage-specific expansion and contraction during evolution. Interestingly, the distribution of these genes was uneven across the chromosomes, with all CYP76 genes being absent on chromosomes 6, 7, and 11 in the seven *Oryza* genus (Figure 1a). CYP76s from the seven *Oryza* genus could be divided into three groups according to the phylogenetic analysis and multiple sequence alignment, and Group 1 contained the largest number of CYP76 genes (Figure 1b). Gene structure analysis revealed that the number of exons and introns ranged from one to four, suggesting that CYP76s may gain or lose exons or introns because of chromosomal rearrangements (Figure 2d). Collinearity analysis revealed that all gene pairs were generated by SD/WGD or tandem duplication, indicating that SD/WGD and tandem duplication may play important roles in the expansion of the CYP76 gene family in the AA genome *Oryza* genus (Figure 4). Additionally, the Ka/Ks values of all duplication gene pairs ranged from 0.27 to 0.67, which is less than one, suggesting that these genes have undergone strong purifying selection during evolution. Interestingly, the SD/WGD duplication events were estimated to occur between 49.01 and 88.64 Mya, while most tandem duplication events were estimated to occur between 11.64 and 28.37 Mya, indicating that gene pairs in SD/WGD duplication events were conserved during evolution (Table 1). Furthermore, most of the amino acid sequences of the CYP76 proteins ranged from 443 (ORUFI10G03760.1) to 701 (OBART01G19690.1). However, the amino acid sequences of OBART09G12160.1, ORUFI01G30870.1, ORUFI03G10820.1, and ORUFI10G03800.1 were 1042aa, 994aa, 984aa, and 827aa, respectively, probably because these proteins contain more repetitive motifs relative to other CYP76 proteins. Additionally, most CYP76 proteins (80%) have an isoelectric point greater than seven, suggesting that CYP76 proteins may be basic proteins (Appendix A).

The *CYP76* gene family is known to play important roles in plant growth and abiotic stress response [14,15,16]. Specifically, *AtCYP76C2*, *AtCYP76C5*, and *AtCYP76C6* have been shown to be highly expressed in both leaves and flowers in Arabidopsis thaliana [30]. In rice, *OsCYP76-6*, *OsCYP76-14*, *OsCYP76-16*, and *OsCYP76-17* belong to Group 2 in the *CYP76* family and were the orthologs of *AtCYP76C2*, *AtCYP76C5*, and *AtCYP76C6*, and these genes were also highly expressed in leaves, suggesting their involvement in leaf development regulation. Additionally, *AtCYP76C4* was found to be highly expressed in roots according to the TAIR database. *OsCYP76-1*, *OsCYP76-10*, *OsCYP76-13*, and *OsCYP76-16* belong to Group 1 in the *CYP76* family and were the orthologs of *AtCYP76C4*, exhibiting high expression in roots. Furthermore, *AtCYP76C1*, *AtCYP76C2*, *AtCYP76C3* and *AtCYP76C7* were up-regulated under cold, salt, and drought stresses according to the TAIR database, and most *OsCYP76* genes were shown to be similarly up-regulated under these stress conditions (Figure 5).

*O. sativa*, japonica and *O. sativa*, indica are representative crops of cultivated rice in Asia, and the analysis of tolerance differences among subspecies is important for the discovery of tolerance-related genes. Expression differences analysis of the CYP76 gene family between *O. sativa*, japonica and *O. sativa*, indica revealed that the expression levels in roots differed significantly between the two subspecies after cold stress (Figure 6). Moreover, after drought stress, the expression of most CYP76s increased in leaves of *japonica*, while half of them increased and the other half decreased in indica (Figure 6). After salt treatment, we observed increased CYP76 genes expression in leaves of japonica and up-regulation of half of the genes and down-regulation of the other half in leaves of indica (Figure 6). Additionally, most CYP76 genes showed little difference in expression in roots of japonica, whereas in indica, the majority exhibited increased expression under flooding stress (Figure 6). These results revealed functional differentiation within the gene family during evolution and that the CYP76 gene family may be the key genes responsible for the difference in tolerance between indica and japonica.

## 4. Materials and Methods

### 4.1. Plant Material, Growth, and Abiotic Stress Conditions

Rice plants (Nipponbare, *Oryza. Sativa*) were cultivated in the 26 °C growth chamber of Wuhan University, the light/dark photoperiod was 16/8 h, and relative humidity was 60% [27]. Plants were exposed to abiotic stresses at three-leaf stage. To examine the response of the *CYP76s* in *O. sativa*, japonica and *O. sativa*, indica to abiotic stresses, plants at three-leaf stage were subjected to diverse treatments such as 30% polyethylene glycol (PEG 6000), 300 mM sodium chloride (NaCl), flooding, and 5 low temperatures. The treated rice leaves and roots were harvested at time 24 h after abiotic stresses. All the obtained samples were immediately stored at −80 °C until further use. Three biological replicates for each treatment were conducted.

### 4.2. Identification and Phylogenetic Analysis of CYP76 Genes in Rice

The whole genome datasets of *Osj*, *Osi*, *Or*, *Oglu*, *Ogla*, *Om*, and *Ob* were downloaded from Ensembl Plants (http://plants.ensembl.org/index.html, accessed on 1 December 2022). The HMM (Hidden Markov Model) profile of the Cytochrome P450 (PF00067) was obtained from InterPro (https://www.ebi.ac.uk/interpro/, accessed on 1 December 2022) and used as the query to search against the seven species using the HMMER 3.3.2, respectively [31,32]. Meanwhile, sixteen *OsCYP76* protein sequences from the NCBI (https://www.ncbi.nlm.nih.gov/, accessed on 1 December 2022) were downloaded as query sequences to search against the local proteins database through the BLASTP program (E-value of e^−5^). Then, the results of the BLASTP and HMMER methods were intersected [33]. Subsequently, all candidate CYP76 protein sequences were submitted to InterPro databases (https://www.ebi.ac.uk/interpro/search/sequence/, accessed on 20 December 2022) and the NCBI (http://www.ncbi.nlm.nih.gov/cdd/, accessed on 20 December 2022) to test for the inclusion of the CYP76 structural domain and to remove the missing structural domain sequences. The identified protein sequences of CYP76 were subjected to multiple sequence alignments using the default parameters of Clustalw [34]. Subsequently, an unrooted phylogenetic tree was constructed using the neighbor-joining method with a bootstrap value of 1000 times, implemented in MEGA-X software [35]. The resulting tree was further refined using the ITOL online tool. (https://itol.embl.de/, accessed on 20 December 2022) [36]. All accession names of CYP76 genes are shown in Appendix A.

### 4.3. Sequence and Protein Properties of CYP76 Genes in Rice

To conduct a comprehensive analysis of the physical and chemical properties of CYP76 proteins, the number of amino acids, isoelectric point, and molecular weight were analyzed by TBtools [37]. Furthermore, a combination of Cell-Ploc 2.0 (http://www.csbio.sjtu.edu.cn/bioinf/Cell-PLoc-2/, accessed on 20 December 2022) and PSORT (https://www.psort.org/index.html, accessed on 20 December 2022) was used to predict the subcellular localization of these CYP76 proteins to ensure the accuracy of prediction, and MEME (http://meme-suite.org/tools/meme, accessed on 20 December 2022) was employed to identify conserved motifs among these proteins with a maximum of 20 motifs and default parameters [38,39,40]. Additionally, the 2 kb sequences upstream of the transcription start codon of *CYP76* genes were extracted by TBtools and further detected by the PlantCARE [41] (http://bioinformatics.psb.ugent.be/webtools/plantcare/html/, accessed on 20 December 2022). The information on cis-elements is shown in Appendix A.

### 4.4. Chromosomal Localization, Orthogroup Identification, and Collinearity Analysis

To gain insight into the genomic organization of CYP76 genes, the information on their chromosomal positions and gene structures was extracted from GFF3 files, which was further visualized by TBtools. To explore the duplication events of CYP76s, the collinearity analysis was performed by MCScanX with default parameters, and the distribution of collinear pairs was presented by TBtools [42]. The information on duplication gene pairs is shown in Appendix A. Furthermore, the TBtools was applied to calculate nonsynonymous (Ka), synonymous (Ks), and Ka/Ks ratios of duplicate genes and divergence time (T) was calculated by T = Ks/(2 × 9.1 × 10^−9^) × 10^−6^ Mya program [43]. Additionally, to analyze the ortholog gene groups of CYP76 genes, an all-vs-all BlastP search was generated by the Diamond software, and the results were the input file for OrthoFinder v2.5.4 software [44,45]. The information on Or groups is shown in Appendix A. The information on duplication gene pairs is shown in Appendix A.

### 4.5. Expression Analysis of OsCYP76 Genes

To study the expression profiles of OsCYP76s, tissues at different developmental stages were retrieved from the RiceXPro database (https://ricexpro.dna.affrc.go.jp/, accessed on 20 December 2022), and those in salt, drought, flooding, and cold stress conditions datasets were retrieved from the TENOR database (https://tenor.dna.affrc.go.jp/, accessed on 20 December 2022) and visualized using TBtools [31,37,46].

### 4.6. RNA Extraction and qPCR Analysis

Total RNA from the samples were extracted by TRIzol reagent. The first-strand cDNA was synthesized using HiScript III 1st Strand cDNA Synthesis SuperMix for qPCR (Cat No.11141ES60; Yeasen, Shanghai, China), and quantitative analysis was performed using Hieff UNICON Universal Blue qPCR SYBR Green Master Mix (Cat No.11184ES08; Yeasen, Shanghai, China) according to the manufacturer’s protocol. The untreated samples were used as the control group. Three technical replicates were performed for each gene. The qRT-PCR reaction was performed on a CFX96 TouchTM Real-Time PCR Detection System (Bio-Rad, Hercules, CA, USA) with the following thermal cycling conditions: pre-denaturation at 95 °C for 10 min, denaturation at 95 °C for 10 s, annealing at 60 °C for 10 s, and extension at 72 °C for 15 s. The reactions were repeated for 40 cycles, and solubility curves were plotted from 65 °C to 95 °C to detect primer specificity. The actin gene (UBI) was used as an internal reference control, and the primers used in this study were designed by Primer Premier 5.0 (Appendix A). The relative expression level was calculated based on three biological replicates using the 2^−∆∆CT^ method [47].

## 5. Conclusions

In this study, we identified the CYP76 subfamily in seven AA genome species by conducting a comprehensive genome-wide analysis and a rigorous phylogenetic analysis to infer their evolutionary relationships. We then studied their chromosomal localization, orthogroup identification, and collinearity. Moreover, we examined the expression patterns of these genes in different tissues under various abiotic stresses in *O. sativa*, japonica and *O. sativa*, indica. In summary, our results provide valuable insights into the functional diversity and evolutionary history of the CYP76 subfamily in rice and may pave the way for the development of new strategies for improving stress tolerance and agronomic traits in rice.

## Figures and Tables

**Figure 1 ijms-24-08522-f001:**
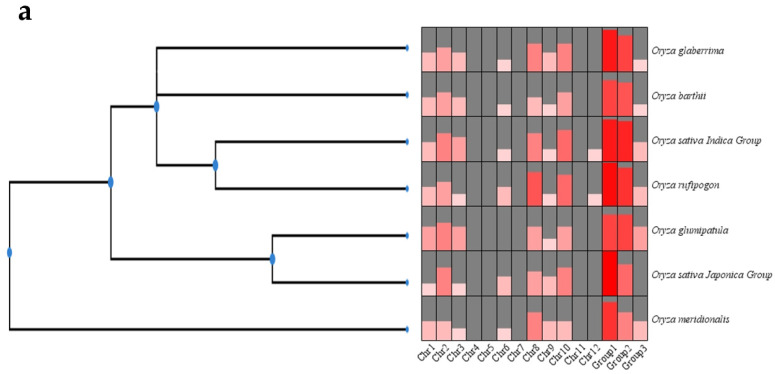
CYP76 proteins identified in the seven *Oryza*. (**a**) Comparison of the number of *CYP76* gene family members in the seven *Oryza*. (**b**) Phylogenetic tree of CYP76s in the seven *Oryza*.

**Figure 2 ijms-24-08522-f002:**
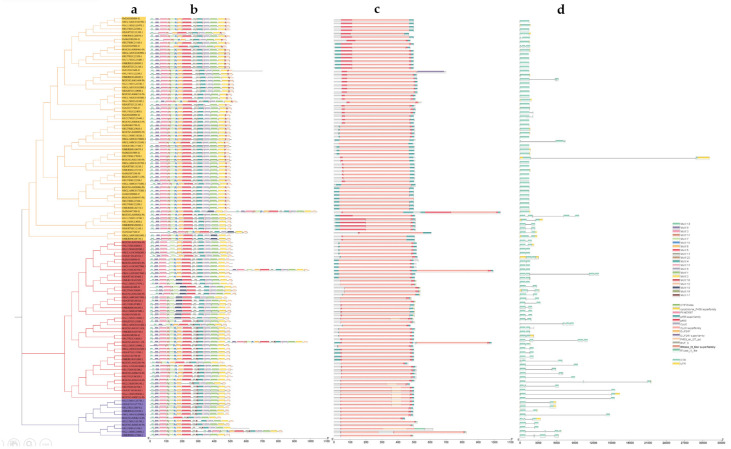
Phylogenetic relationship, conserved motifs, functional domains, and gene structure of CYP76 subfamily. (**a**) Phylogenetic tree of CYP76 genes in the seven *Oryza*. (**b**) The distributions of conserved motifs. (**c**) Conserved functional structural domains of *CYP76s*. (**d**) The structure of exons and introns of CYP76 genes.

**Figure 3 ijms-24-08522-f003:**
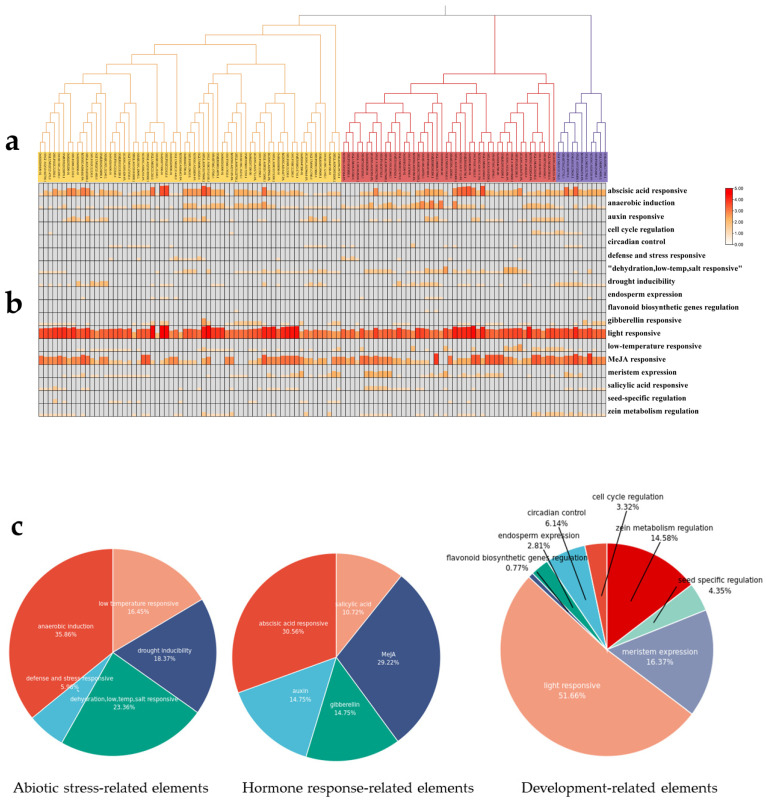
Cis-elements analysis of CYP76 genes: (**a**) Phylogenetic tree of CYP76 genes in the seven *Oryza*. (**b**) Heatmap of variety cis-elements in CYP76 genes (**c**) Cis-elements distribution of CYP76 genes in abiotic stress-related elements, hormone response-related elements, and development-related elements.

**Figure 4 ijms-24-08522-f004:**
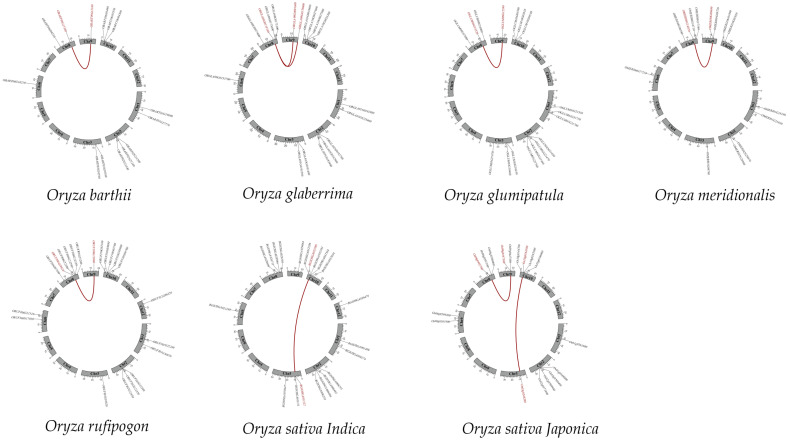
Collinearity analysis of CYP76 genes in the seven *Oryza*.

**Figure 5 ijms-24-08522-f005:**
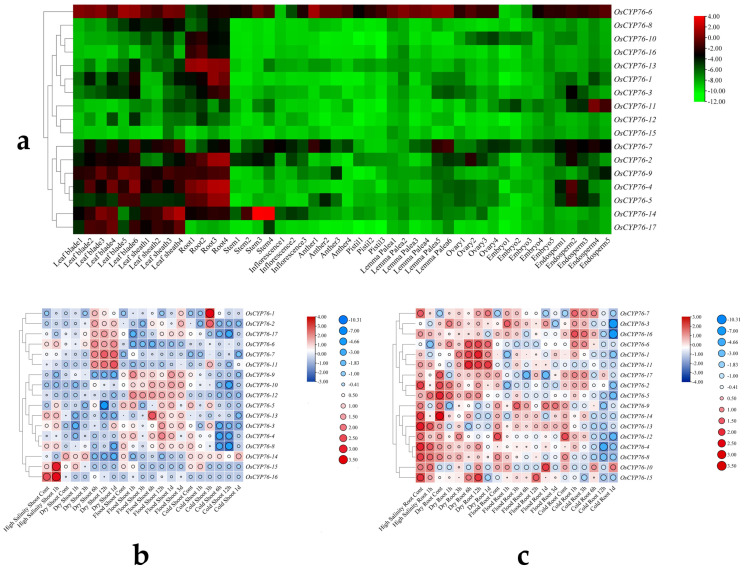
Expression pattern analysis of *OsCYP76* genes and RNA-seq analysis under abiotic stresses. (**a**) Expression pattern analysis of *OsCYP76s* in various developmental stages. (**b**,**c**) RNA-seq analysis of *OsCYP76s* under salt, drought, flooding, and cold stresses in shoot and root, respectively.

**Figure 6 ijms-24-08522-f006:**
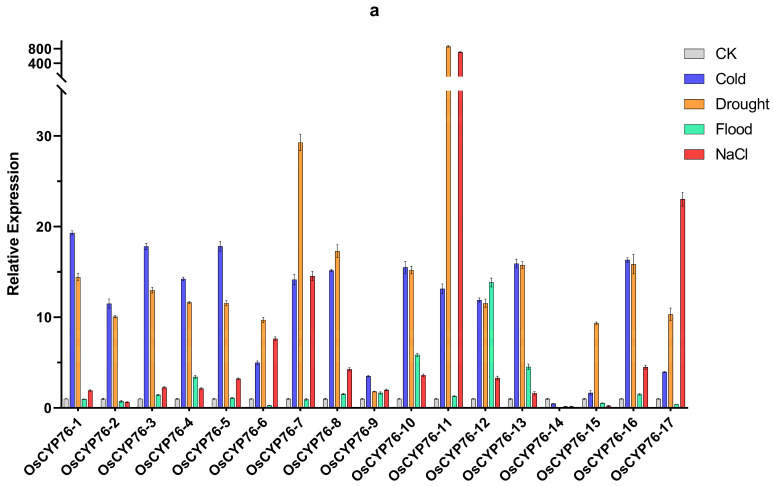
Relative expression levels of CYP76 genes under abiotic stresses in *O. sativa*, japonica and *O. sativa*, indica. (**a**,**b**) Relative expression levels of CYP76 genes under cold, drought, flooding, and salt stresses in *O. sativa*, japonica leaves and roots, respectively. (**c**,**d**) Relative expression levels of CYP76 genes under cold, drought, flooding, and salt stresses in *O. sativa*, indica leaves and roots, respectively.

**Table 1 ijms-24-08522-t001:** The Ka/Ks and T values of duplication gene pairs in *Oryza*.

Oryza	Gene Pairs	Ka/Ks	Date (Mya)	Type of Selection	Type of Duplication
*O. sativa*, *japonica*	*Os10t0164500-00/Os03t0248200-01*	0.33	70.73	Purifying	SD/WGD
	*Os08t0465700-01/Os09t0447300-01*	0.58	53.93	Purifying	SD/WGD
	*Os02t0569000-01/Os02t0569400-01*	0.62	28.38	Purifying	Tandem
	*Os09t0447300-01/Os09t0447500-01*	0.62	50.43	Purifying	Tandem
*O. sativa*, *indica*	*BGIOSGA032580-PA/BGIOSGA011112-PA*	0.30	88.64	Purifying	SD/WGD
	*BGIOSGA011112-PA/BGIOSGA011111-PA*	0.49	22.41	Purifying	Tandem
*O. rufipogon*	*ORUFI08G19610.1/ORUFI09G13060.1*	0.58	53.97	Purifying	SD/WGD
	*ORUFI08G22280.1/ORUFI08G22290.1*	0.47	16.07	Purifying	Tandem
*O. meridionalis*	*OMERI08G14070.1/OMERI09G09690.1*	0.62	51.28	Purifying	SD/WGD
*O. glumaepatula*	*OGLUM08G18320.1/OGLUM09G12590.1*	0.60	53.01	Purifying	SD/WGD
	*OGLUM01G31750.1/OGLUM01G31760.1*	0.22	63.73	Purifying	Tandem
	*OGLUM02G21470.1/OGLUM02G21480.1*	0.29	11.64	Purifying	Tandem
*O. glaberrima*	*ORGLA08G0153100.1/ORGLA09G0091600.1*	0.67	49.01	Purifying	SD/WGD
	*ORGLA08G0153100.1/ORGLA09G0170600.1*	0.62	50.55	Purifying	SD/WGD
	*ORGLA02G0182700.1/ORGLA02G0182800.1*	0.27	12.99	Purifying	Tandem
	*ORGLA03G0101400.1/ORGLA03G0101500.1*	0.48	23.09	Purifying	Tandem
*O. barthii*	*OBART08G17500.1/OBART09G12160.1*	0.58	53.54	Purifying	SD/WGD
	*OBART03G10580.1/OBART03G10590.1*	0.47	23.33	Purifying	Tandem

## Data Availability

Data are contained within the article or Appendix A.

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
