# Peer review of "Genome-Wide Identification and Evolution Analysis of the CYP76 Subfamily in Rice (Oryza sativa)"

_ijms, 2023, doi:10.3390/ijms24108522_

Round 1

Reviewer 1 Report

Some papers published by the journal have the very same structure: Introduction (I), Results (R), Discussion (D), Materials and Methods (M), and Conclusion (C).  Nevertheless, it would be easier to understand and also to argue the discussion using the classical structure IMRDC.

In the pdf file in the attachment are signed some errors, typing problems, and notes to the manuscript.

Some figures like  Fig.2, Fig.5, and Fig.6 in the format presented are virtually unreadable.

The references  47, 48 and 49 in my view are out of order of citation. 

Author Response

Thank you for your letter and the reviewers' comments on our manuscript. These comments are valuable and helpful for us to revise and improve our paper, and they are also important guidance for our research. We have carefully studied these comments and made revisions, which we hope will be approved by all of you.

Response to Reviewer 1 Comments

Point 1: Some papers published by the journal have the very same structure: Introduction (I), Results (R), Discussion (D), Materials and Methods (M), and Conclusion (C).  Nevertheless, it would be easier to understand and also to argue the discussion using the classical structure IMRDC.

Response 1: Thank you for your suggestion, this article is modified according to the organizational knot of the journal IJMS.

Point 2: In the pdf file in the attachment are signed some errors, typing problems, and notes to the manuscript..

Response 2: Thank you very much for your patience in commenting, I have finished the revision work. And fixed my other mistakes according to your correction. Eg. page 1, line number 21.

Point 3: Some figures like  Fig.2, Fig.5, and Fig.6 in the format presented are virtually unreadable.

Response 3: Thank you for your comments, I will re-upload a higher resolution image.

Point 4: The references 47, 48 and 49 in my view are out of order of citation.

Response 4: Thanks to your comments, I have rechecked these references.

Reviewer 2 Report

Dear Authors

The manuscript is well written and I have one suggestion regarding the real time RT-PCR experiments and the presentation of results as graphs. In the present form, the graphs are not informative. The graphs can be split for different species of Oryza or in any other convenient way so as it represents the gene expression pattern in a more understandable manner. 

Thank you

The manuscript needs minor editing of language and typographical errors eg.page 13, line number 352.

Author Response

Thank you for your letter and the reviewers' comments on our manuscript. These comments are valuable and helpful for us to revise and improve our paper, and they are also important guidance for our research. We have carefully studied these comments and made revisions, which we hope will be approved by all of you.

Response to Reviewer 2 Comments

Point 1: The manuscript is well written and I have one suggestion regarding the real time RT-PCR experiments and the presentation of results as graphs. In the present form, the graphs are not informative. The graphs can be split for different species of Oryza or in any other convenient way so as it represents the gene expression pattern in a more understandable manner.

Response 1: Thanks to your comments, I redrew the image of qpcr. The redrawn image can clearly reflect the difference of CYP76 among subspecies

Point 2: The manuscript needs minor editing of language and typographical errors eg.page 13, line number 352.

Response 2: Thank you for your comments, I have reworked the sentence.eg. page 13, line numbe 359.

Reviewer 3 Report

To,

The Chief Editor,

IJMS, MDPI,

Manuscript ID: ijms-2361688

Subject: Submission of comments on the manuscript in “IJMS"

Dear Chief Editor IJMS, MDPI,

Thank you very much for the invitation to consider a potential reviewer for the manuscript (ID: ijms-2361688). My comments responses are furnished below as per each reviewer’s comments. 

Dear Chief Editor,

In the reviewed manuscript, the authors conducted a genome-wide analysis of CYP76 subfamily in seven AA genome species: Oryza sativa ssp. japonica, Oryza sativa ssp. indica, Oryza rufipogon, Oryza glaberrima, Oryza meridionalis, Oryza barthii and Oryza glumaepatula were identified and classified into three groups, and it was found that group1 contains the largest numbers of members. The gene duplication analysis revealed that the CYP76 subfamily expanded mainly in the form of SD/WGD and underwent strong purifying selection during evolution. We further examined the expression patterns of these genes in different tissues under various abiotic stresses in O. sativa, japonica and O. sativa, indica and revealed functional differentiation within the gene family during evolution and CYP76 gene family may be the key genes responsible for the difference in tolerance between indica and japonica. In conclusion, our results provide valuable insights into the functional diversity and evolutionary history of the CYP76 subfamily and pave the way for the development of new strategies for improving stress tolerance and agronomic traits in rice. However, in my opinion, the MS needs major revisions. I have some suggestions to improve this manuscript: 

  1. I have read the entire manuscript and my initial comment is that manuscript is poorly written. I have significant concerns about the grammar and vocabulary of the manuscript; therefore, I recommend the authors to used an English proofreading service.
  2. In the manuscript title, the Oryza sativa must be italic.

3.    Abstract does not reflect the whole story, revise it. The abstract should highlight the most important results of the parameters and characteristics assayed.

  1. Keyword must in alphabtical order.

5.    The content of the introduction is effective in essence but very poorly presented, significant improvements are needed in presenting the proper background of the work undertaken..

  1. The figures are quite low resolution and difficult to make out. Higher-resolution versions will be needed for publication. Further, text in figure is not readble. for example, in Figures 1A, 1B, 2, 3, 4, 5, and 6.
  2. In Material and Methods:- indicate how many replicates assayed in each analysis/parameter. The number of samples or biological and technical replicates should be mentioned for each parameter in the methods.
  3. qRT-PCR methodology provided is also very vague and confusing. Please provide more details like what was the calibrator used in the study. I assume the authors have used the control as the calibrator. If so, the authors should not include the control within the bar graph as it represents the fold change between the treated vs control and a fold change of “1” for the ‘control’ doesn’t make any sense.  Also, would be good to provide details on what reagents (details of probes used, if any, if SYBR was used then details for that, etc.) and real time PCR machine were used in the current study.
  4. The discussion should be interpreted with the results as well as discussed in relation to the present literature.  
  5. Conclusion section is very lengthy. The author should emphasize this in a better way.
  6. References: shall have to correct the whole References according to the ”Instructions for the Authors”, e.g. title should not be in italics, the Journal name is in italics, and the author shall have to use the abbreviated name Journals cited the year must be bold, the scientific name must be italics etc. Moreover, duplication of reference numbering. Please check all references carefully.

Best wishes and thank you

To,

The Chief Editor,

IJMS, MDPI,

Manuscript ID: ijms-2361688

Subject: Submission of comments on the manuscript in “IJMS"

Dear Chief Editor IJMS, MDPI,

Thank you very much for the invitation to consider a potential reviewer for the manuscript (ID: ijms-2361688). My comments responses are furnished below as per each reviewer’s comments. 

Dear Chief Editor,

In the reviewed manuscript, the authors conducted a genome-wide analysis of CYP76 subfamily in seven AA genome species: Oryza sativa ssp. japonica, Oryza sativa ssp. indica, Oryza rufipogon, Oryza glaberrima, Oryza meridionalis, Oryza barthii and Oryza glumaepatula were identified and classified into three groups, and it was found that group1 contains the largest numbers of members. The gene duplication analysis revealed that the CYP76 subfamily expanded mainly in the form of SD/WGD and underwent strong purifying selection during evolution. We further examined the expression patterns of these genes in different tissues under various abiotic stresses in O. sativa, japonica and O. sativa, indica and revealed functional differentiation within the gene family during evolution and CYP76 gene family may be the key genes responsible for the difference in tolerance between indica and japonica. In conclusion, our results provide valuable insights into the functional diversity and evolutionary history of the CYP76 subfamily and pave the way for the development of new strategies for improving stress tolerance and agronomic traits in rice. However, in my opinion, the MS needs major revisions. I have some suggestions to improve this manuscript: 

  1. I have read the entire manuscript and my initial comment is that manuscript is poorly written. I have significant concerns about the grammar and vocabulary of the manuscript; therefore, I recommend the authors to used an English proofreading service.
  2. In the manuscript title, the Oryza sativa must be italic.

3.    Abstract does not reflect the whole story, revise it. The abstract should highlight the most important results of the parameters and characteristics assayed.

  1. Keyword must in alphabtical order.

5.    The content of the introduction is effective in essence but very poorly presented, significant improvements are needed in presenting the proper background of the work undertaken..

  1. The figures are quite low resolution and difficult to make out. Higher-resolution versions will be needed for publication. Further, text in figure is not readble. for example, in Figures 1A, 1B, 2, 3, 4, 5, and 6.
  2. In Material and Methods:- indicate how many replicates assayed in each analysis/parameter. The number of samples or biological and technical replicates should be mentioned for each parameter in the methods.
  3. qRT-PCR methodology provided is also very vague and confusing. Please provide more details like what was the calibrator used in the study. I assume the authors have used the control as the calibrator. If so, the authors should not include the control within the bar graph as it represents the fold change between the treated vs control and a fold change of “1” for the ‘control’ doesn’t make any sense.  Also, would be good to provide details on what reagents (details of probes used, if any, if SYBR was used then details for that, etc.) and real time PCR machine were used in the current study.
  4. The discussion should be interpreted with the results as well as discussed in relation to the present literature.  
  5. Conclusion section is very lengthy. The author should emphasize this in a better way.
  6. References: shall have to correct the whole References according to the ”Instructions for the Authors”, e.g. title should not be in italics, the Journal name is in italics, and the author shall have to use the abbreviated name Journals cited the year must be bold, the scientific name must be italics etc. Moreover, duplication of reference numbering. Please check all references carefully.

Best wishes and thank you

Author Response

Thank you for your letter and the reviewers' comments on our manuscript. These comments are valuable and helpful for us to revise and improve our paper, and they are also important guidance for our research. We have carefully studied these comments and made revisions, which we hope will be approved by all of you.

Response to Reviewer 3 Comments

Point 1: I have read the entire manuscript and my initial comment is that manuscript is poorly written. I have significant concerns about the grammar and vocabulary of the manuscript; therefore, I recommend the authors to used an English proofreading service.

Response 1: Thank you for your suggestions and for pointing out the shortcomings in the article. We have reworked the article and optimized the grammar and vocabulary. We hope to receive your approval.

Point 2: In the manuscript title, the Oryza sativa must be italic.

Response 2: Thank you for your correction, we have revised it. And based on that, other similar errors within the article have been corrected as well.

Point 3: Abstract does not reflect the whole story, revise it. The abstract should highlight the most important results of the parameters and characteristics assayed.

Response 3: Thanks to your careful comments, we have revised the content of the abstract. eg. Line 25.

Point 4: Keyword must in alphabtical order.

Response 4: Thank you for your careful comments, we have revised the order of the keywords. line 36.

Point 5: The content of the introduction is effective in essence but very poorly presented, significant improvements are needed in presenting the proper background of the work undertaken.

Response 5: Thanks to your suggestions, we have revised the content of the introduction and improved the background. Line 49 and 64.

Point 6: The figures are quite low resolution and difficult to make out. Higher-resolution versions will be needed for publication. Further, text in figure is not readble. for example, in Figures 1A, 1B, 2, 3, 4, 5, and 6.

Response 6: Thank you for your suggestion, we are ready to re-upload the images with higher resolution.

Point 7: In Material and Methods:- indicate how many replicates assayed in each analysis/parameter. The number of samples or biological and technical replicates should be mentioned for each parameter in the methods..

Response 7: Thanks to your careful comments, we have added information on biological replicates and technical replicates to the Materials and Methods. Line 363 and 425.

Point 8: qRT-PCR methodology provided is also very vague and confusing. Please provide more details like what was the calibrator used in the study. I assume the authors have used the control as the calibrator. If so, the authors should not include the control within the bar graph as it represents the fold change between the treated vs control and a fold change of “1” for the ‘control’ doesn’t make any sense.  Also, would be good to provide details on what reagents (details of probes used, if any, if SYBR was used then details for that, etc.) and real time PCR machine were used in the current study.

Response 8: Thank you for your careful comments, we have modified the picture to represent it in another way. Line 425 and 258.

Point 9: The discussion should be interpreted with the results as well as discussed in relation to the present literature.

Response 9: Thanks to your suggestion, we have added a section for discussion together with other literature and in the context of the available experimental results. Line 301.

Point 10: Conclusion section is very lengthy. The author should emphasize this in a better way.

Response 10: Thanks to your comments, we have streamlined and revised the conclusion section. Line 434.

Point 11: References: shall have to correct the whole References according to the ”Instructions for the Authors”, e.g. title should not be in italics, the Journal name is in italics, and the author shall have to use the abbreviated name Journals cited the year must be bold, the scientific name must be italics etc. Moreover, duplication of reference numbering. Please check all references carefully.

Response 11: Thanks to your comments, we have revised the references in accordance with Instructions for the Authors.